# Impact of Cell Layout on Bandwidth of Multi-Frequency Piezoelectric Micromachined Ultrasonic Transducer Array

**DOI:** 10.3390/mi16010049

**Published:** 2024-12-31

**Authors:** Wanli Yang, Huimin Li, Yuewu Gong, Zhuochen Wang, Xingli Xu, Xiaofan Hu, Pengfei Niu, Wei Pang

**Affiliations:** 1State Key Laboratory of Precision Measurements Technology and Instrument, Tianjin University, Tianjin 300072, China; 2Precision Measurement Research and Development Center, Nanchang Research Institute, Sun Yat-sen University, Nanchang 330200, China; 3College of Mechanical and Electrical Engineering, Beijing University of Chemical Technology, Beijing 100029, China

**Keywords:** PMUT, multiple frequency, bandwidth, coupling, ultrasound imaging

## Abstract

Piezoelectric micromachined ultrasonic transducers (PMUTs) show considerable promise for application in ultrasound imaging, but the limited bandwidth of the traditional PMUTs largely affects the imaging quality. This paper focuses on how to arrange cells with different frequencies to maximize the bandwidth and proposes a multi-frequency PMUT (MF-PMUT) linear array. Seven cells with gradually changing frequencies are arranged in a monotonic trend to form a unit, and 32 units are distributed across four lines, forming one element. To investigate how the arrangement of cells affects the bandwidth, three different arrays were designed according to the extent of unit aggregation from the same frequency. Underwater experiments were conducted to assess the acoustic performance, especially the bandwidth. We found that the densest arrangement of the same cells produced the largest bandwidth, achieving a 92% transmission bandwidth and a 50% burst-echo bandwidth at 6 MHz. The mechanism was investigated from the coupling point of view by finite element analysis and laser Doppler vibrometry, focusing on the cell displacements. The results demonstrated strong ultrasound coupling in the devices, resulting in larger bandwidths. To exploit the advanced bandwidth but reduce the crosstalk, grooves for isolation were fabricated between elements. This work proposes an effective strategy for developing advanced PMUT arrays that would benefit ultrasound imaging applications.

## 1. Introduction

Ultrasonic-related technology is widely used in various fields, including distance measurement [1], non-destructive testing [2,3], medical imaging [4,5], and ultrasound therapy [6]. Ultrasound imaging plays an important role in medical diagnosis, and ultrasound transducers are the key components of imaging systems. Bulk piezoelectric ceramic ultrasonic transducers manufactured by the mechanical cutting–filling–lapping method are widely used; however, they possess limitations including a large size, high power consumption, and high fabrication costs, especially for high-frequency ones due to the relatively low yield, making them difficult to miniaturize and hindering their portability [7]. Comparatively, micromachined ultrasonic transducers (MUTs) are developed using the semiconductor process and have progressed in various practical applications with the development of micro-electro-mechanical systems (MEMSs). Their good acoustic impedance matching makes them potential alternatives to conventional ultrasound transducers [8,9]. By employing mature integrated circuit (IC) manufacturing technology, MUTs have the advantages of simple fabrication and ease of manufacturing compact large array devices, and their compatibility with complementary metal oxide semiconductors (CMOSs) makes integration with other electronic devices possible [10,11]. Compared to capacitive MUTs (CMUTs), which have a high electromechanical coupling factor and wide bandwidth, PMUTs do not require a high bias voltage, making them safer than CMUTs in portable applications [12,13].

Ultrasound bandwidth is a crucial factor determining the resolution of imaging, while PMUTs always exhibit low bandwidths because they generate acoustic waves based on deformation induced by the e31 mode of piezoelectric materials which lack damping, and the electromechanical coupling factor is typically low, generally not exceeding 10%. Thus, their usage in commercial ultrasound imaging applications is limited. A few works have proposed solutions to increase the bandwidth of PMUTs by filling the cavity with liquid or silicone [14]. However, retaining the air structure on one side of the membrane layer is much more prevalent and common.

Multi-frequency coupling in array designs is feasible for bandwidth expansion [15,16]. In our previous research, we proposed a structural design scheme to increase the bandwidth by adding varied mass loads on PMUT cells of the same size to achieve multi-frequency combinations within one array [17]. The displacement of the designed PMUT array was 1.5 times higher than that of a conventional PMUT array, and the bandwidth reached 29.7%, compared to the 16.8% of the conventional PMUT array. This indicates that multi-frequency combination is an effective method for improving the bandwidth, thereby achieving high axial resolution in imaging.

Generally, in a PMUT device, many primary cells are parallelly connected together to form one unit, making this the basic unit for forming single-element probe, 1D, or 2D arrays. Within a unit of an MF-PMUT array, there are multiple cells with different resonant frequencies. The cells’ resonant frequencies can be altered by adjusting the cavity’s geometry and dimensions. Many researchers also adjust the aspect ratio in rectangular membranes to change the frequency and combine multiple vibration modes to achieve multi-frequency integration. Sadeghpour et al. presented an array composed of five AlN PMUTs of different membrane dimensions, working at frequencies between 540 and 2360 kHz to broaden the bandwidth, with cells spaced 30 μm apart in a tree-like arrangement. However, in pressure measurements, five resonance frequencies were observed, with each frequency having an individual bandwidth of 50–150 kHz. This indicates that the MF-PMUT did not achieve effective coupling to flatten the passband of the frequency response [18]. Cai et al. designed a AlN-based MF-PMUT array, placing 72 cells ranging from 300 μm to 180 μm in an 8-row by 9-column array, and the diameter of the MF-PMUT gradually decreased along the row direction. The resonant frequencies were between 1.6 MHz and 2.2 MHz, and the bandwidth of the array was extended to approximately 65%. Nonetheless, the echo signal spectrum showed several minor peaks, indicating room for improvement in multi-frequency coupling [19]. Sun et al. proposed a multi-frequency PMUT based on the resonant modes of rectangular membranes. By altering the length-to-width ratios of 2, 4, and 6 of the rectangular membranes, different designs and bandwidth expansion results were achieved. However, the experimental results showed that the three designed samples exhibited distinct multiple frequency bands. Large mode frequency gaps and imperfect mode merging caused significant ripples between the frequency bands, leading to poor multi-frequency coupling [20]. In these studies, the frequency spectrum curves of the multi-frequency combination arrays in the experimental results similarly exhibit several distinct resonance peaks with attenuation between each peak. This indicates that the designed resonant frequencies are not well coupled. Moreover, the arrangement of cells and the layout of the unit composed of multiple cells within the array have not been studied, which we believe may affect the multi-frequency coupling effect and thereby impact the device bandwidth.

In this study, we integrated seven PMUT cells with varying resonance frequencies into a linear array to expand the bandwidth. The resonant frequency of each cell was varied by changing the transverse dimension of the PMUT diaphragm. Cells of different sizes were arranged in a frequency sequence to form a unit, and multiple units were connected in parallel to form an element. We propose three arrangements and study the closeness of the cells in the whole element. Our study focuses on the impact of the layout of units in the element on the device bandwidth. Finite element simulations are used to explain the facilitating effect of coupling between cells on multi-frequency combinations from the perspective of center displacement, followed by acoustic experiments. The experimental results reveal that the closer the cells of similar frequencies are arranged, the higher the bandwidth of the device. The design and study of PMUT arrays presented in this paper will provide guidance for the research of wide-bandwidth multi-element PMUT arrays.

## 2. Finite Element Analysis and Fabrication of PMUT Device

### 2.1. Modeling

The individual PMUT cell in the simulation was designed as the structure shown in Figure 1, with key geometric parameters listed in Table 1. The material of the substrate is Si, and the cavity depth is about 1.2 μm. The thickness of the Mo layer, acting as the top and grounded bottom electrodes, is approximately 100 nm, and the AlN piezoelectric layer between the two electrodes is about 500 nm thick. The diameter of the top electrode is 70% of the cavity size to achieve the optimal electromechanical coupling factor [16]. The outermost layer is approximately 1.2 μm thick SiO_2_, serving as an elastic layer that makes the membrane vibrate in flexible mode. To avoid mesh partition errors caused by some overly small domains, the bottom electrode radius in the model was appropriately increased to 30 μm.

In the above membrane stack structure, we aimed to design a PMUT device with a center frequency of about 6 MHz. The resonant frequency is determined as the frequency when each PMUT cell achieves its maximum displacement. Finite element analysis simulation and theoretical calculations show that when the cavity diameters are 38 μm, 40 μm, 42 μm, 44 μm, 46 μm, 48 μm, and 50 μm, the resonance frequencies in air are 12.9 MHz, 11.7 MHz, 10.8 MHz, 9.9 MHz, 9.1 MHz, 8.4 MHz, and 7.8 MHz, respectively, and the resonance frequencies in water are 7.6 MHz, 7 MHz, 6 MHz, 5.6 MHz, 5.2 MHz, 4.4 MHz, and 4.2 MHz, respectively. These cells were arranged in parallel with a space of 75 μm, following the layouts shown in Figure 2a–c, and we called the corresponding arrays A-type, B-type, and C-type arrays.

In order to investigate the impact on the bandwidth of different cell couplings, a 3D model of a 2 × 14 PMUT cell combination was established. This 2 × 14 PMUT cell combination will be repeated in the final manufactured array, so we used this specific combination for simulation. The space between cells was set to 75 μm, and a cylindrical water region with a diameter of 6 mm and a height of 6 mm was added outside the combination to simulate the working environment. The boundary of the model was set as a perfectly matched layer (PML) to absorb the sound wave. The substrate and each PMUT cell were assigned solid mechanics. For each cell, the electrodes and piezoelectric layer were set to the electrostatic domain, and the boundary condition of the cavity was set as free. The interface between the solid and water domains was set as an acoustic–structure boundary, with the remaining boundaries fixed. The above settings for the simulation model are shown in Figure 3a. Each layer used finer free triangular meshes at the cell positions, while the remaining areas used coarser free triangular meshes, and the outer water region was meshed with a sweep of 160 elements, as shown in Figure 3b,c.

### 2.2. Simulation of the Impact of the Layout on the Cells’ Interaction

To study the acoustic output, the displacement of cells under different frequencies was simulated. A 1 V voltage excitation was applied to the top electrode, which means the cells were excited in parallel, with the parameterized frequency sweep range set to 3–10 MHz for the frequency domain simulation. The operational status of the cells was judged by observing the central displacement, where maximum displacement indicated resonance. Figure 4a–c show the cells’ displacement at a 4 MHz excitation frequency. Taking 4 MHz as an example, at this frequency, cells with a cavity radius of 25 μm were nearly resonant, while other cells remained non-resonant. We found that cells with the same resonant frequency differed in maximum amplitude, and non-resonant cells also exhibited certain displacements. Under different arrangements, cells with the same resonant frequency showed different displacements, and the displacement ratio between resonant and non-resonant cells also varied.

To further quantify this, we extracted the frequencies and displacements of these cells when resonant. For frequency analysis, we compared the frequencies of each cell in the three models with the simulated result of a single cell under the same conditions, as shown in Figure 5. The resonant frequencies of the seven cells in the single-cell simulation were 7.6 MHz, 7 MHz, 6 MHz, 5.6 MHz, 5.2 MHz, 4.4 MHz, and 4.2 MHz, respectively. In the C-type combination, each cell’s frequency closely matched that of a single cell, with minimal deviation. However, cells in the A-type and B-type combinations exhibited greater frequency deviations relative to that of a single cell.

For displacement analysis, we extracted the displacements from a row of 14 cells when cells with different sizes were under resonant conditions. Due to the different arrangements of cells, to better observe the displacement ratio of non-resonant cells compared to resonant cells, we sorted these displacements in descending order. Taking cells with resonant frequencies of about 4 MHz, 6 MHz, and 8 MHz in resonance as examples, the descending displacement plots of the 14 cells are shown in Figure 6a–c, respectively, and the patterns for other sizes of resonant cells are similar to those in Figure 6. The displacements of two resonant cells under three arrangements are highlighted by the orange boxes. As shown in the figure, the non-resonant cell displacements are similar; however, under these conditions, the resonant cells’ displacements are the highest in the C-type combination and the lowest in the B-type combination. Taking Figure 6a as an example, in the A-type, B-type, and C-type combinations, the percentage of displacement of non-resonant cells compared to resonant cells is less than 48%, 64%, and 25%, respectively. These percentages are 80%, 89%, and 56% and 77%, 80%, and 66% in Figure 6b,c, respectively. This suggests that the C-type combination would achieve a better amplitude, but may be unbalanced in the frequency domain. On other hand, the mutual influence between cells in the B-type combination is stronger, and cells with different resonant frequencies may couple better in the frequency response, resulting in a higher bandwidth.

Based on the frequency and displacement simulation results, in the B-type array with the most densely arranged cells of the same frequency, the frequency deviation is the largest among the various cell sizes, and the resonant cell displacements vary the most due to the non-resonant cells’ influence. Thus, it can be inferred that the closer the arrangement of cells with the same resonant frequency, the stronger the coupling between them, while a staggered arrangement weakens this interaction. This inference will be further validated through subsequent experiments.

### 2.3. Fabrication Based on Simulation

Optical images of the A-type, B-type, and C-type multi-frequency PMUT arrays were captured by a microscope camera (Leica ICC50 W, Leica Microsystems, Wetzlar, GER), as shown in Figure 7a–c. The cross-section of a cell captured by scanning electron microscopy (SEM) is shown in Figure 7d,e. Each element contains 4 columns, with each column comprising 8 units, resulting in a total of 224 cells integrated into a single element. Variations in cavity sizes can be visually distinguished by the sizes of the top electrodes in the optical image. Since cells with the same resonant frequency are rarely placed together inside columns directly connected by Au, we only etched isolation grooves between each column to suppress the crosstalk. This not only reduced the complexity of the design and manufacturing but also ensured certain isolation. We anticipated that this frequency combination would meet our design objective. The devices were fabricated using semiconductor techniques and the sacrificial layer process. Detailed fabrication steps can be found in our previous publications [21].

## 3. Results and Discussion

### 3.1. Frequency Characteristics

The electrical impedance and phase of the three PMUT arrays in air were measured using an impedance analyzer (E4990A, Keysight, CA, USA), as depicted in Figure 8a–c. These figures illustrate the electrical signals of the multi-frequency cells of the PMUT arrays, with the seven narrow peaks in the phase spectrum corresponding to the resonant frequencies of the seven PMUT cells with different cavity sizes. It can be seen that the resonant frequencies of the cells in the three arrays are almost identical. Manufacturing errors during the manufacturing process or other mechanical factors which may affect the actual cavity size can lead to a certain deviation between the measured resonant frequencies and the simulated resonant frequencies. The effective electromechanical coupling factor $k_{eff}^{2}$ can be calculated by Formula (1):(1)$$k_{eff}^{2}=\frac{f_{a}^{2}-f_{r}^{2}}{f_{a}^{2}}$$
where $f_{r}$ is the electrical resonance frequency and $f_{a}$ is the anti-resonance frequency. Since it is challenging to determine the high-frequency impedance peaks, the electromechanical coupling factors corresponding to the first three resolvable impedance peaks of the A-type, B-type, and C-type devices were evaluated to be about 0.76%, 1.17%, and 0.66%; 0.66%, 1.17%, and 1.08%; and 0.76%, 0.71%, and 0.87%, respectively. Among these three designs, the B-type device has a better overall electromechanical conversion capability, which may lead to better performance.

At high resonance frequencies, the phase peaks are also not very pronounced. Therefore, we took the relatively more distinct C-type array as an example, and a laser Doppler vibrometer (LDV, OFV 512 and OFV3001, Polytec, Baden-Württemberg, Germany) was used to further measure the actual resonant frequencies of cells with varying cavity sizes, which were 12.723 MHz, 11.663 MHz, 10.7 MHz, 9.859 MHz, 9.093 MHz, 8.439 MHz, and 7.865 MHz. These were compared with the resonant frequencies obtained from the finite element simulations of single cells and showed good agreement (Figure 8d).

### 3.2. Acoustic Performance

To verify the underwater acoustic performance of the multi-frequency coupled PMUT array, several experiments were conducted to evaluate the echo performance and transmission performance. A 15 μm thick polydimethylsiloxane (PDMS) film (SC15, Westru Technology, Hangzhou, China) was attached to the surface to protect it from water ingress into the cavity, and it also acted as a passivation layer. This process was accomplished by plasma cleaning and treating the surfaces of the device with PDMS to introduce hydroxyl groups, enabling their bonding.

#### 3.2.1. Transmission Performance

We operated the PMUT array in transmission mode and performed a frequency sweep of the input sine wave signal from 3 MHz to 10 MHz, with a step of 0.1 MHz, a cycle count of 4, and an input amplitude of 60 Vpp. We aligned the needle hydrophone (NH2000, Precision Acoustics, Dorchester, UK) with the PMUT array at a distance of about 1 cm, and the position was adjusted to maximize the amplitude of the received signal, which was displayed on the oscilloscope (RTB2002, Rohde & Schwarz, München, Germany) after a 21 dB amplification by the hydrophone. The received signals of the A-type, B-type, and C-type arrays were compared (Figure 9).

We recorded the amplitude corresponding to each frequency and plotted the data as a curve, as shown in Figure 10. The center frequency is around 6.3–6.4 MHz, and the −6 dB bandwidth of the B-type array can reach 92%, while that of the A-type array is 79% and that of the C-type array is 91%. It is worth noting that the curve of the C-type array shows some attenuation between the two frequency band peaks.

#### 3.2.2. Burst-Echo Test

A burst-echo test was conducted on the PMUT array, where it was excited by a single 6 MHz sinusoidal pulse with a 60 Vpp amplitude generated by a multi-element signal generator, lasting approximately 160 ns. The water–air interface at a distance of about 2.5 cm was used as the reflective surface, with the PMUT array acting as both the transmitter and receiver. The time-domain signal was filtered, collected, and amplified through a pulser-receiver (DPR500, Imaginant, New York, USA) and displayed on the oscilloscope (RTB2002, Rohde & Schwarz, München, GER). The corresponding spectrum was calculated using the fast Fourier transform (FFT), as shown in Figure 11. The center frequency is about 6 MHz, and the −6 dB bandwidth of the B-type array under one-cycle burst excitation can reach 50%, while that of the A-type array is 33%. The C-type array shows significant attenuation around 5.5 MHz, with two peaks appearing in the spectrum, and the bandwidths of the two frequency band peaks are about 24% and 26%, respectively. Table 2 presents a comparison between the performances observed in other studies and this work on bandwidth expansion by multi-frequency design.

From the above echo and transmission test results, it can be concluded that the B-type array, where cells of the same frequency are arranged most densely, has the highest bandwidth, followed by the A-type array, while the C-type array, with staggered units, shows two peaks in the spectrum with a clear attenuation between the two narrow frequency bands. This suggests that a higher density of cells with the same resonant frequency may enhance multi-band coupling in multi-frequency arrays, potentially increasing the bandwidth. We speculate that this may be due to the mutual interactions between cells, with cells of the same resonant frequency enhancing vibrations, while interactions between cells of different frequencies may inhibit vibrations. To achieve a broader bandwidth in a multi-frequency array, the B-type design, where cells with identical resonance frequencies are arranged more closely, should be recommended.

### 3.3. Cells’ Interaction Measured by LDV

To further assess the interaction between different elements of the PMUT array, we also used an LDV (OFV 512 and OFV3001, Polytec, Baden-Württemberg, GER) to measure the surface vibration amplitudes of the cells. In our design, each element consists of four columns of cells connected to the same Au pad. During the measurement, we applied a continuous 5 Vpp periodic sine wave excitation to one element, while the other elements remained unexcited, as shown in Figure 12.

By measuring the vibration amplitudes of the cells in the two neighbor elements, we can assess the impact of the crosstalk signals transmitted from the excited element. The laser was focused on the center of the cell in the excited element for measurement, and the amplitude of vibration was displayed as an electrical signal on an oscilloscope (RTB2002, Rohde & Schwarz, München, GER). By adjusting the frequency of the input signal frequency, the amplitude of the received signal also varied, with the maximum amplitude indicating that the cell was in its strongest vibrational resonance state. This allowed us to determine the resonant frequency of each cell. The resonant frequencies of the four columns of cells to be excited were confirmed, and the arithmetic mean was taken as the excitation frequency for subsequent measurements of that element to ensure that the cells in the excited element were as close to resonance as possible.

#### 3.3.1. Crosstalk Comparison Between Arrays with Different Cell Arrangements

The normalized vibration displacements of the excitation element and neighbor elements of the A-type, B-type, and C-type PMUT arrays are shown in Figure 13. It can be seen that the B-type array, which concentrates cells with the same cavity size, exhibits the most significant crosstalk, while the A-type and C-type arrays, which have a certain staggered arrangement, show relatively lower and comparable crosstalk. It can be observed that the excitation consistency of the A-type array is inferior to that of the C-type array, yet it still exhibits a considerable crosstalk effect, suggesting that the C-type array has lower crosstalk.

Thus, it is evident that a denser arrangement of cells with the same frequency leads to stronger coupling between cells, while more staggered arrangements of cells with the same frequency tend to reduce it. This result is also consistent with the conclusions of the simulation.

#### 3.3.2. Crosstalk Reduction

Crosstalk between cells in an MF-PMUT array can, to some extent, strengthen the coupling of cells with different resonant frequencies, thereby expanding the bandwidth. However, excessive crosstalk might lead to unintended responses in some elements under non-excited conditions due to the influence of other elements. To achieve physical isolation, we etched grooves between each column, with a width of 10 μm. The etching depth of the grooves may influence the isolation effectiveness differently, so we designed three grooves with different depths of 2 μm, 5 μm, and 20 μm.

The normalized vibration displacements of excitation elements with grooves of varying etching depths, compared to the neighbor elements, are illustrated in Figure 14. The results show that etching 2 μm deep grooves that only separate the vibration layer yielded the least effective reduction in crosstalk, while arrays with 5 μm grooves that were etched down to the Si substrate exhibited lower crosstalk, with about −30 dB. Arrays etched with deeper 20 μm grooves showed a comparable crosstalk reduction to the 5 μm grooves. It can be concluded that employing isolation grooves to separate both the vibration layer and the cavity can effectively reduce crosstalk, and deeper grooves do not lead to a more substantial reduction in crosstalk effects. For our designs, the optimal groove depth is 5 μm.

While etching grooves reduces crosstalk, we needed to determine whether the bandwidth of the array will be affected or not. We selected the B-type array as an example to compare the bandwidths of arrays with three different groove depths. A 15 μm thick PDMS film was adhered to the surface to protect it against moisture ingress into the cavity. Burst-echo tests were conducted, and the water–air interface served as the reflection surface. The time-domain signals were collected, with the corresponding frequency spectrum calculated based on the FFT, as shown in Figure 15.

From the spectrum curves, it can be observed that the three devices with varying groove depths have similar bandwidths. This demonstrates that the design of the isolation grooves does not significantly affect the central frequency and bandwidth. Therefore, after the introduction of the grooves, the coupling between cells in the same column, which reflects the bandwidth of the device, was not affected, while the crosstalk between columns was diminished.

Therefore, to exploit the advantages of the advanced bandwidth but reduce crosstalk caused by cells’ coupling, we propose to increase the coupling within one channel but form trenches for isolation between different channels in an array.

## 4. Conclusions

This study focused on the impact of the specific layout of multi-frequency cells on the bandwidth improvement of a PMUT array. A new MF-PMUT array was proposed and fabricated, integrating cells with seven resonant frequencies in the same array to expand the bandwidth. These seven cells form a unit and are arranged with their frequencies changing gradually in a monotonic trend. We designed three arrays based on the different arrangements of units, according to the extent of unit aggregation from the same frequency in one ultrasound element.

The multi-frequency characteristics and underwater acoustic performance were evaluated. The underwater acoustic test results demonstrate that the bandwidth was successfully broadened. We found that the device with the units and their symmetry combined to more densely arrange cells of the same size achieved a wider bandwidth than the other two. Simply repeating a unit within an element results in a narrower bandwidth, while interleaving the unit and its symmetry to less densely arrange cells of the same size leads to a spectrum with two split peaks.

Based on both the finite element analysis and LDV testing, we found that arrangements that achieved a higher bandwidth showed stronger coupling between cells, which could enhance the bandwidth. However, this strong coupling is mainly due to acoustic field/vibration coupling rather than mechanical crosstalk transmitted through the membrane, since we found that etching isolation grooves did not reduce the bandwidth. In practical multi-element PMUT array design, we suggest that etching isolation grooves between cells within an element is unnecessary, and etching isolation grooves between different elements can effectively reduce crosstalk.

## Figures and Tables

**Figure 1 micromachines-16-00049-f001:**
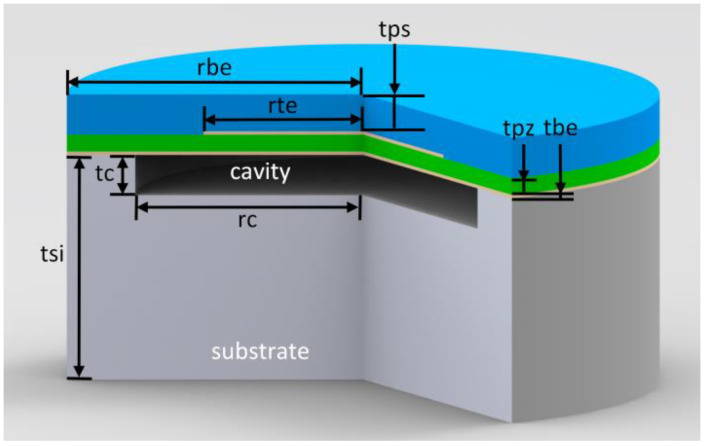
Schematic diagram of the structure of a single cell and the geometric parameters of its simulation model.

**Figure 2 micromachines-16-00049-f002:**
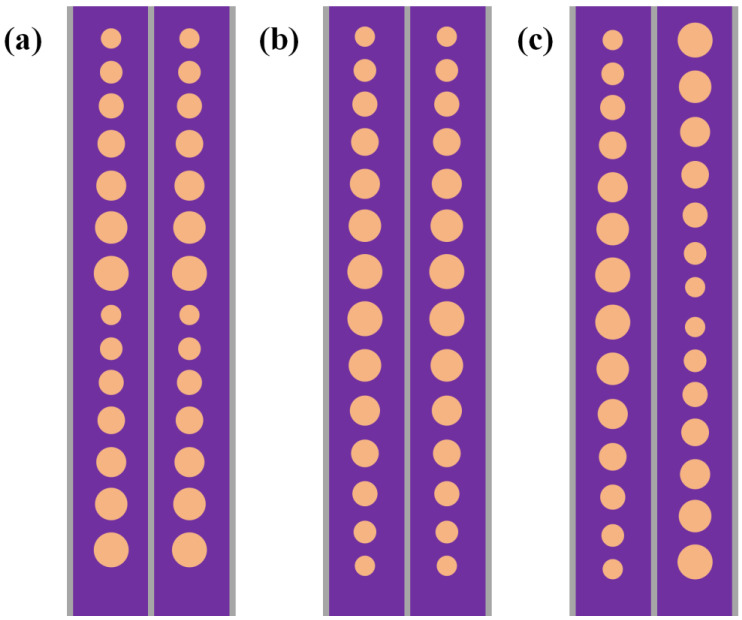
Arrangement (**a**–**c**) of cells with different resonant frequencies.

**Figure 3 micromachines-16-00049-f003:**
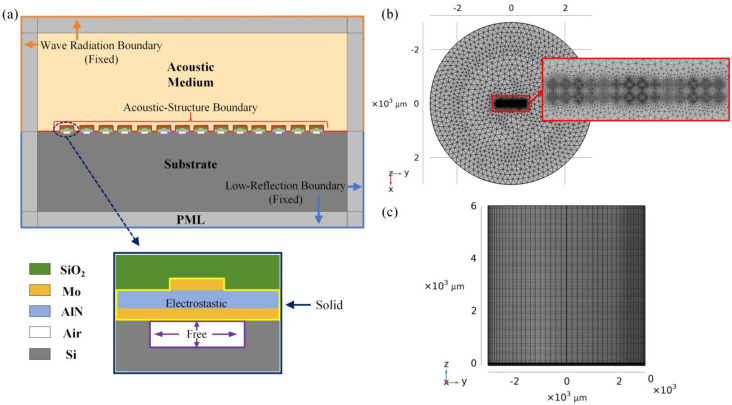
(**a**) The FEA model for the PMUT array, displaying boundary conditions. The model is not depicted in proportion to show the configuration details. Simulation model mesh examples of the B-type combination: (**b**) free triangular mesh of the combination and (**c**) global mesh sweep.

**Figure 4 micromachines-16-00049-f004:**
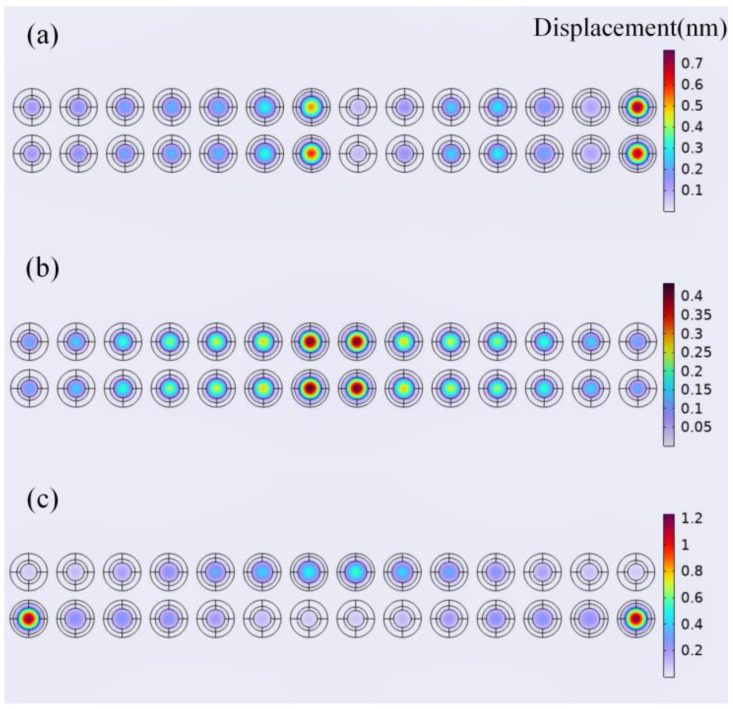
Displacement of the (**a**) A-type, (**b**) B-type, and (**c**) C-type combinations under a 1 Vpp excitation at a frequency of 4 MHz applied to the top electrode.

**Figure 5 micromachines-16-00049-f005:**
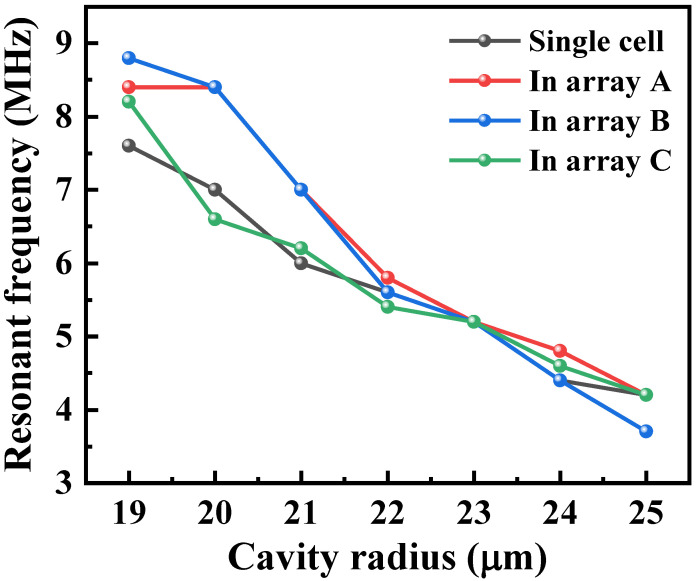
Resonant frequencies of cells corresponding to various cavity sizes in the simulation of a single cell and the A-type, B-type, and C-type combinations.

**Figure 6 micromachines-16-00049-f006:**
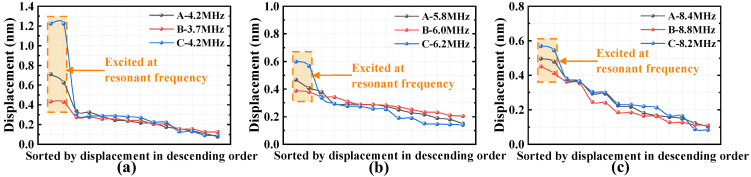
Displacement of each cell: (**a**) when the cell with a resonant frequency of about 4 MHz is in resonance; (**b**) when the cell with a resonant frequency of about 6 MHz in resonance; (**c**) when the cell with a resonant frequency of about 8 MHz in resonance.

**Figure 7 micromachines-16-00049-f007:**
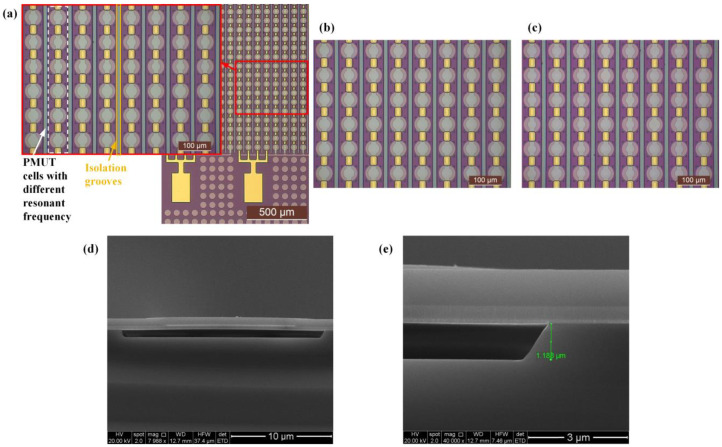
Optical images of (**a**) A-type, (**b**) B-type, and (**c**) C-type multi-frequency PMUT arrays. (**d**) A cross-section of a cell and (**e**) its enlarged view at the edge of the cavity.

**Figure 8 micromachines-16-00049-f008:**
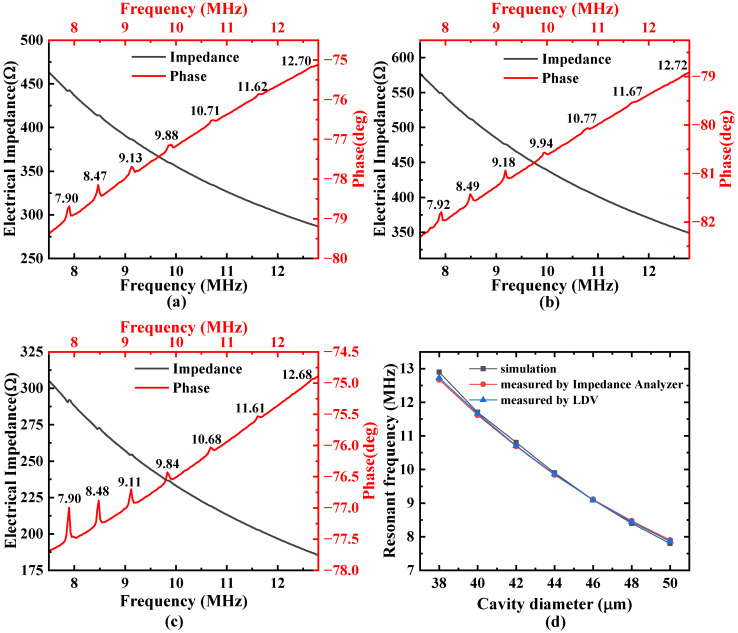
Impedance and phase characteristics of multi-frequency PMUT arrays under the (**a**) A-type, (**b**) B-type, and (**c**) C-type designs; (**d**) comparison schematic of actual measured resonant frequency and simulation results.

**Figure 9 micromachines-16-00049-f009:**
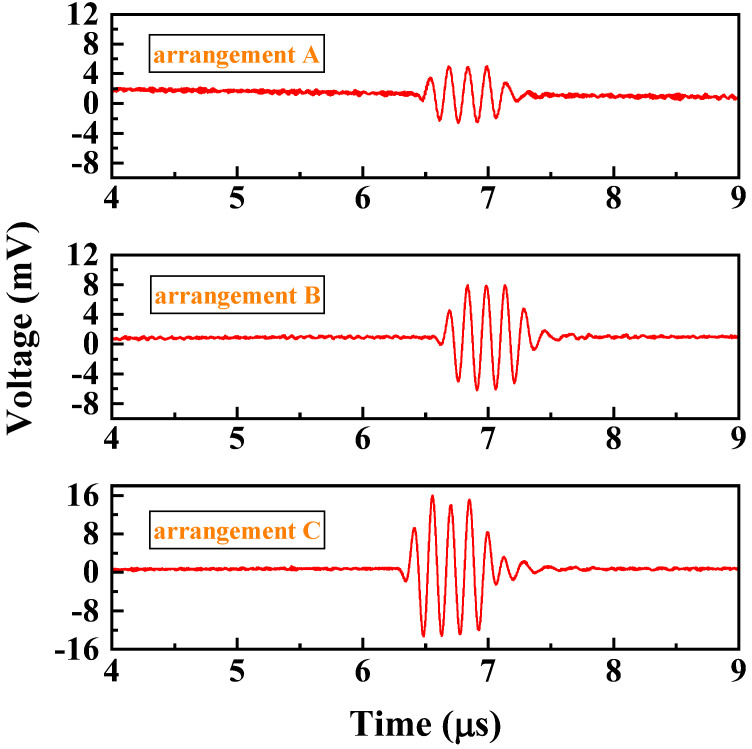
Waveform of the signal received by the hydrophone from the PMUT array transmission.

**Figure 10 micromachines-16-00049-f010:**
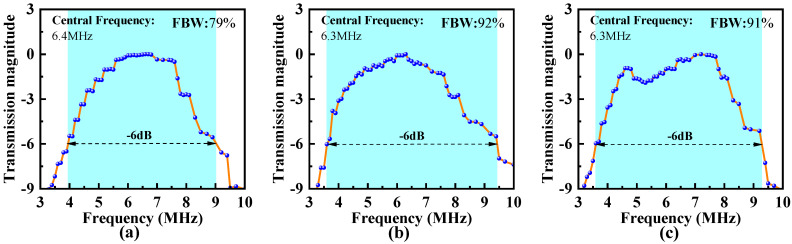
The amplitude–frequency variation curve of the received signal during the frequency sweep transmission of the (**a**) A-type, (**b**) B-type, and (**c**) C-type PMUT array.

**Figure 11 micromachines-16-00049-f011:**

Time-domain response signals and their spectra of the 1-cycle burst echo of the (**a**) A-type, (**b**) B-type, and (**c**) C-type PMUT arrays under single-pulse sine wave excitation.

**Figure 12 micromachines-16-00049-f012:**
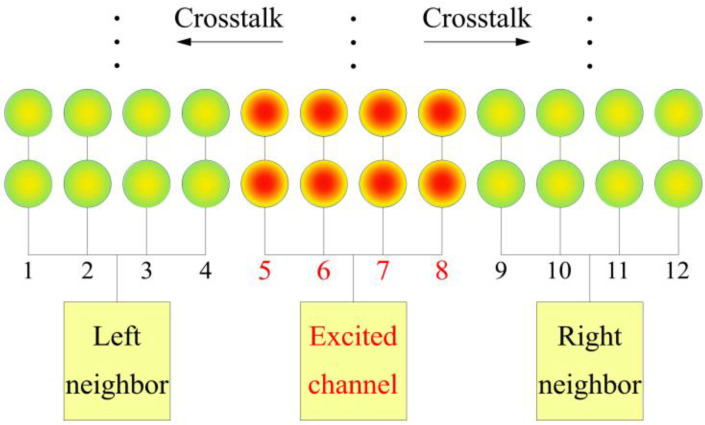
Diagram of the use of an LDV to measure crosstalk.

**Figure 13 micromachines-16-00049-f013:**
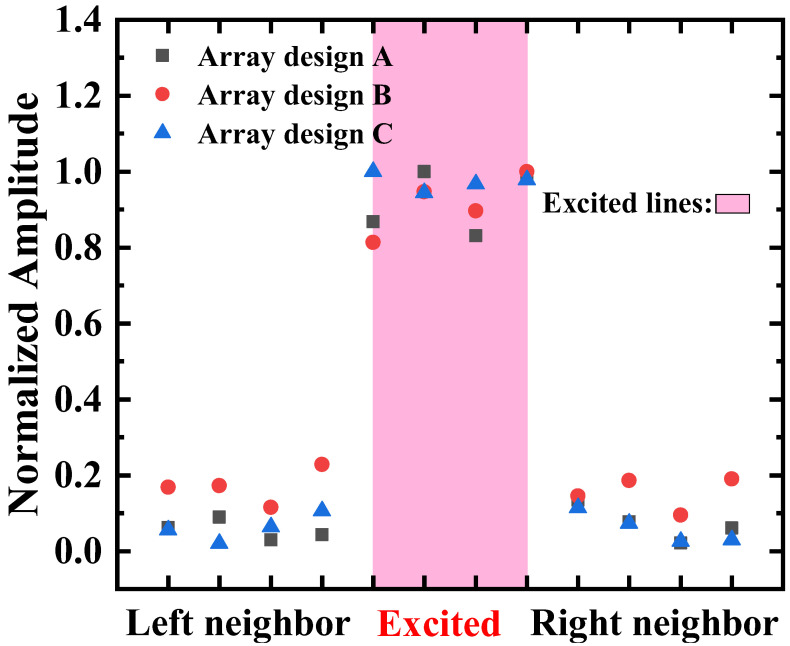
Comparison of normalized vibration displacement between excitation elements and their neighbor elements: comparison of arrays with different arrangements.

**Figure 14 micromachines-16-00049-f014:**
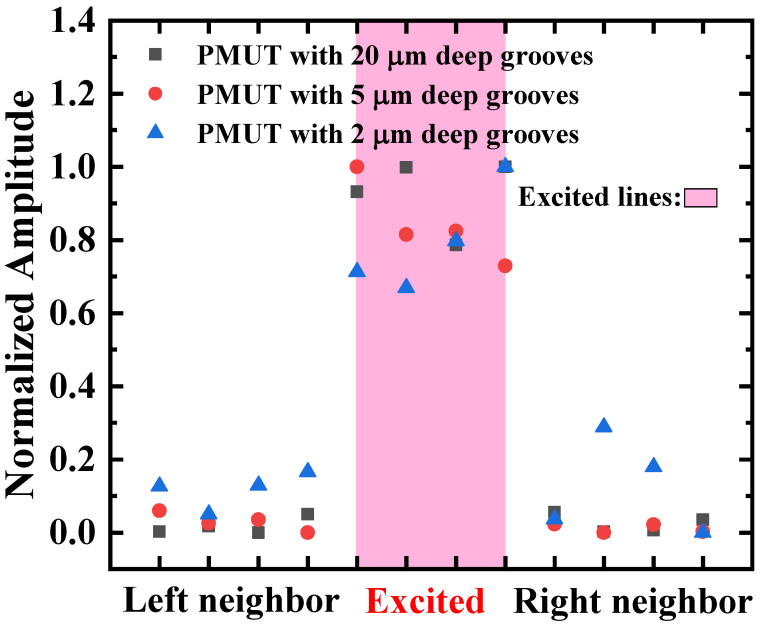
Comparison of normalized vibration displacement between excitation elements and their neighbor elements: comparison of arrays with isolation grooves of different depths.

**Figure 15 micromachines-16-00049-f015:**
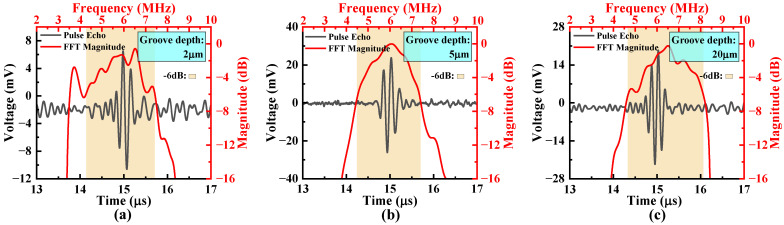
Time-domain response signals and their FFT spectra of PMUT arrays with isolation grooves of different depths under single-pulse sinusoidal signal excitation: (**a**) grooves etched to a depth of 2 μm; (**b**) grooves etched to a depth of 5 μm; (**c**) grooves etched to a depth of 20 μm.

**Table 1 micromachines-16-00049-t001:** Geometric parameters of the simulation model.

Structure	Material	Thickness (μm)	Radius (μm)
Elastic layer	Silicon dioxide	t_ps_ 1.2	r_be_ 30
Top electrode	Molybdenum	t_te_ 0.1	r_te_ (0.7r_c_) 17.5, 16.8, 16.1, 15.4, 14.7, 14, 13.3
Piezoelectric layer	Aluminum nitride	t_pz_ 0.5	r_be_ 30
Bottom electrode	Molybdenum	t_be_ 0.1	r_be_ 30
Cavity	Vacuum	t_c_ 1.2	r_c_ 25, 24, 23, 22, 21, 20, 19
Substrate	Silicon	t_si_ 100	r_be_ 30

**Table 2 micromachines-16-00049-t002:** Comparison of bandwidths of MF-PMUT arrays in different studies.

Study	Method	Number of Frequency Bands *	Central Frequency	−6 dB Bandwidth
Luo et al. [5]	Six sparse spiral combinations with the same cells in a Fermat spiral array	1	840 kHz when transmitting	108%
Sadeghpour et al. [18]	Five different cells in a tree-like structure	5	about 150 kHz, 500 kHz, 750 kHz, 1 MHz, and 1.5 MHz when transmitting	each with 50–150 kHz individual bandwidth (multiple bands)
Cai et al. [19]	Four different cells in a linear array	1	2 MHz in pulse echo	65%
Sun et al. [20]	Rectangular membrane with multiple vibration modes	2	0.685 MHz and 1.415 MHz when transmitting	118% and 36% (two bands)
This work	Seven different cells in a linear array	1	6.3 MHz in transmission and 6 MHz in 1-cycle burst-echo test	92% in transmission and 50% in 1-cycle burst-echo test

* If multiple peaks appear in the frequency response and there is significant attenuation between the peaks, multiple bands will be observed. This indicates insufficient band coupling.

## Data Availability

The original contributions presented in this study are included in the article; further inquiries can be directed to the corresponding author.

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
