# Peer review of "Impact of Cell Layout on Bandwidth of Multi-Frequency Piezoelectric Micromachined Ultrasonic Transducer Array"

_micromachines, 2024, doi:10.3390/mi16010049_

Round 1
Reviewer 1 Report
Comments and Suggestions for Authors
Comments to authors,
In the manuscript titled as “Impact of cell layout on the bandwidth of multi-frequency piezoelectric micromachined ultrasonic transducer array”, Yang et al. presented a MF-PMUT array with 7 resonant frequencies in the same array to expand the bandwidth. And, designed three arrays based on the different arrangement of units, according to the extent of unit aggregation from the same frequency in one ultrasound element. Underwater acoustic test results demonstrated that the bandwidth has been successfully broadened.
This manuscript could be published, yet, the following issues must be addressed before its publication.
1. What instrument was used to capture the optical image shown in Figure 7? The measuring scale should be provided for the size reference.
2. The three arrays based on the different arrangement of units designed in this article should all be displayed in optical images. And, suggest the author to add a cross-sectional view of the PMUT.
3. The electrical impedance and phase of the three PMUT arrays in this article should all be displayed in Figure 8 (a).
4. How to deposit a 15 μm-thick polydimethylsiloxane (PDMS) film on the surface of the PMUT array? Author should add a description of the process in manuscripts.
5. Why are the data in Figure 11 (b) and Figure 15 (b) completely consistent? Even if the same device undergoes the same test, similar data can be obtained by multiple measurements, and completely consistent data should not appear.
In summary, this manuscript still has some technical problems. Major revision should be made.
Reviewer 2 Report
Comments and Suggestions for Authors
The authors propose a multi-frequency piezoelectric micro-machined ultrasonic transducer (PMUT) for increased bandwidth in an array transducer. Bandwidth expansion is achieved by using a multi-cell design with different cell radii, resulting in varying resonance frequencies. The linear array PMUTs have been designed and realized. Both the transmission and the transmission-reception bandwidths are characterized. This study is undoubtedly interesting. I recommend publication in MDPI Micromachines, provided the authors make the following improvements.
1. In the introduction, the authors state that MEMS MUTs have advantages over traditional ultrasound transducers and that PMUTs have a narrower frequency bandwidth compared to CMUT devices. However, this raises several fundamental questions: Can MEMS piezo e31 flexure mode devices match conventional e33 thickness resonance transducers in terms of their electromechanical coupling factor and potentially replace them? Does the high bias voltage required for CMUTs pose a real safety concern in applications, despite offering superior performance? Additionally, what explains the lower bandwidth of PMUTs compared to CMUTs, given that both utilize the same vibration structure, namely a clamped membrane on a silicon wafer?
2. The description of the design and the finite element model is unclear, particularly the terminology used for "cell" and "array" (line 130). It should be specified whether the array is composed of 2x14 PMUT cells. The term "pitch" is ambiguous—does it refer to the spacing within a cell or across the array (line 124)? The rationale for using a cylindrical water environment with a diameter of 6mm and a height of 6mm as the working environment is also unclear. How a PMUT cell is substrated? The authors need to provide a detailed description of the array's geometry, including exact boundaries and boundary conditions in the finite element modeling.
3. The clarity of how the resonance frequencies are defined needs improvement. It should be explained what the resonance of the cells in the array signifies (line 154), and how this differs from the resonance of a single cell (line 155). Additionally, it must be clarified whether all the cells are electrically excited in parallel.
4. In section 3.2.1, the authors measured the transmission bandwidth by sweeping the frequency with a burst of three sinusoidal periods (line 234). Observations from the last signal presented in Figure 9 indicate that such a short duration of the burst (narrow band) does not suffice to ensure a stable amplitude response from the PMUT output, thereby questioning the accuracy of the measurement.
5. In section 3.2.2, the authors utilized wideband excitation to measure the return frequency response. However, an excitation using a one-cycle burst (line 262) does not constitute a wideband signal. The results obtained do not adequately represent the response spectrum and typically require normalization with the excitation spectrum.
6. What is the reason for measuring the transmission frequency response at a 1 cm distance from the PMUT (line 235), while the PMUT is placed at 2.5 cm from the water reflection surface for measuring the return frequency response (line 256)?
Reviewer 3 Report
Comments and Suggestions for Authors
The work explores three PMUT array designs including cells with different sizes, aiming to achieve wider bandwidth by merging different resonant modes. Many aspects of this paper are unclear, especially concerning the design choices, the measurement data processing, and the ultimate goal of the performed work.
Here are my main concerns:
1) More information about the model size and applied boundary conditions are needed to make proper considerations about the acoustic waves propagating in the simulated volume before drawing conclusions from the bandwidth results. From Fig. 3, it seems that the propagating medium volume is not much greater than the transducer element size, hence, the waves propagating in the coupled medium are not plane, unless some assumptions were made at the borders. Is the model accounting for diffraction effects? Are there reflections from the fluid boundaries that affect the transmission and reflection transfer functions computation? Do the authors use PML to absorb the radiated pressure field? Answers to these questions should also be supported by pressure field plots.
2) Concerning Section 2, Fig. 4 does not provide much information about the vibrational behaviour of cells. Of course, cells with fundamental resonant frequency closer to the exemplary frequency chosen (i.e. 4 MHz) will have greater displacement than those that naturally resonate at different frequency. Different displacements observed in cells with same geometrical features are related to their location in the array, since they experience different mechanical boundary conditions and crosstalk effect. The discussion about this phenomenon is mostly limited to the computation of the percentage of displacement of non-resonant cells compared to resonant cells, thereby lacks a theoretical background or explanation. Therefore, the comments do not provide great insight into the phenomenon, that remains largely unexplored. Crosstalk is addressed also in the Discussion section; however, no theoretical explanation is provided alongside the observation of the phenomenon.
3) In addition, the simulated displacement results are not compared to experiments. Conversely, the pressure measured by the hydrophone presented in Section 3 was not simulated. In general, a direct comparison between simulation and experiments is not possible, except for the resonant frequency values. Therefore, the reason for including both simulation results and measurement results in the work, that cannot be compared for validation, is unclear.
4) Concerning Section 3, the electrical impedance measurement data in Fig. 8 should be presented in a way that allows assessing clearly the impedance values and the phase peaks for each resonant mode, for instance by showing the results in separated plots of smaller frequency spans.
5) Furthermore, in line 241, the statement about “unexpected and incoherent signal” should be clarified. What is unexpected, and why? What is the signal uncoherent with?
6) For what concerns the burst-echo tests, was the signal compensated for the effects of propagation (nonlinearities introduced by air)? Was it normalized by dividing by the input signal (i.e. is it a proper transfer function that can be compared with other results in the literature, or is it simply the measured signal?)? This is especially important because the results are directly compared with those published in other works, but it is not stated clearly if the quantities are measured in the same operating conditions.
Overall, it is unclear what objective this study wants to pursue: is it the improvement of multifrequency designs (if so, the design choices leading to the three configurations investigated – A , B, C – should also be discussed)? Is it the reduction of crosstalk by mechanical measures, like application of isolation trenches? Is it the resonant frequency uniformity assessment? Or all of these? Ultimately, what is the relevance of the findings? How does this study improve the understanding and advances PMUT research?
Reviewer 4 Report
Comments and Suggestions for Authors
Piezoelectric micromachined ultrasonic transducers (PMUTs) show considerable promise for application in ultrasound imaging, while the limited bandwidth of the traditional PMUTs largely affects the imaging quality. This paper focuses on how to arrange cells with different frequencies to maximize the bandwidth and proposes a multi-frequency PMUT (MF-PMUT) linear array. 7 cells with frequencies changing gradually are arranged by a monotonic trend to form a unit. And 32 units are distributed at 4 lines forming one element. To investigate how the arrangement of cells affects bandwidth, three different arrays were designed, according to the extent of unit aggregation from the same frequency. Underwater experiments were conducted to assess the acoustic performance, especially bandwidth. We find the densest arrangement for the same cells produces the largest bandwidth, achieving 92% transmission bandwidth and 50% burst-echo bandwidth at 6 MHz. The mechanism is investigated from the coupling point of view by finite element analysis and laser Doppler vibrometer about the cell displacements. The results demonstrate stronger ultrasound coupling in devices resulting in larger bandwidths. To exploit advanced bandwidth but reduce crosstalk, grooves for isolation were fabricated between elements. This work shows an effective strategy for developing advanced PMUT arrays that would benefit ultrasound imaging applications.
In the manuscript, the authors presented arranging cells with different frequencies to maximize the bandwidth and proposed a multi-frequency PMUT (MF-PMUT) linear array. The densest arrangement for the same cells produces the largest bandwidth with 92% transmission bandwidth and 50% burst-echo bandwidth at 6 MHz, which are good results and might be attractive to the relevant fields. However, there are some concerns that need to be solved before publishing.
1. why authors made a 1.2 μm cavity depth for the single-cell element? To increase bandwidth?? Do authors have any simulation results about it?? How to confirm the depth of the 1.2 μm.
2. What are rbe, tbe, rte, tsi short for? Reviewer might know as thickness of piezo or Silicone, but audience might not know. Please clarify this.
3. Do authors add any acoustic matching layers?
4. In Fig. 2, why the circles (piezo) are not the same size? To meet impedance match?? How to align?
5. In Fig 7, what is the substrate for the array? How to layout the copper traces for each element?
6. Fig 8a has lots of peaks for phase, and fr and fa. How to confirm the central frequency??
7. How to get the bandwidths in Fig 10, by using FFT? Why there are dots in the bandwidth?
8. Fig 11c 's bandwidth looks not good.
Comments on the Quality of English Language
minor revision
Round 2
Reviewer 1 Report
Comments and Suggestions for Authors
The authors revised the paper, and the quality of the paper has been improved. The paper could be published.
Author Response
Thank you very much for your careful review and constructive suggestions to our manuscript. We appreciate your affirmation to our revision.
Reviewer 2 Report
Comments and Suggestions for Authors
The authors' response to comment 2 with regard to finite element modeling is incomplete; the problem is not with the boundaries of the water cylinder, but with the need to provide more detail on the boundaries of the PMUT device itself shown in Fig. 3, i.e., the boundaries of the cell and array structures, including mechanical, acoustic, and electrical boundaries. The results of the FEA should be utilized to compare with the measurements reported below, such as electrical impedance (Fig. 8), vibrational displacements, and crosstalk with deep groove structures (Fig. 14), and the differences between types A, B, and C should be cross-referenced.
Reviewer 4 Report
Comments and Suggestions for Authors
The authros have solved most of the questions, it can be accepted after minor revision.
1. Please include the electromechanical coefficient equation and calculate all the related values.
Comments on the Quality of English Language
minor revision
